# Precision prevention in worksite health–A scoping review on research trends and gaps

**Filip Mess** *, **Simon Blaschke***, **Teresa S. Schick**, **Julian Friedrich** *

Technical University of Munich, TUM School of Medicine and Health, Munich, Germany

* filip.mess@tum.de (FM); simon.blaschke@tum.de (SB); julian.friedrich@tum.de (JF)

## Abstract

### Objectives

To map the current state of precision prevention research in the workplace setting, specifically to study contexts and characteristics, and to analyze the precision prevention approach in the stages of risk assessment/data monitoring, data analytics, and the health promotion interventions implemented.

### Methods

Six international databases were searched for studies published between January 2010 and May 2023, using the term "precision prevention" or its synonyms in the context of worksite health promotion.

### Results

After screening 3,249 articles, 129 studies were reviewed. Around three-quarters of the studies addressed an intervention (95/129, 74%). Only 14% (18/129) of the articles primarily focused on risk assessment and data monitoring, and 12% of the articles (16/129) mainly included data analytics studies. Most of the studies focused on behavioral outcomes (61/160, 38%), followed by psychological (37/160, 23%) and physiological (31/160, 19%) outcomes of health (multiple answers were possible). In terms of study designs, randomized controlled trials were used in more than a third of all studies (39%), followed by cross-sectional studies (18%), while newer designs (e.g., just-in-time-adaptive-interventions) are currently rarely used. The main data analyses of all studies were regression analyses (44% with analyses of variance or linear mixed models), whereas machine learning methods (e.g., Algorithms, Markov Models) were conducted only in 8% of the articles.

### Discussion

Although there is a growing number of precision prevention studies in the workplace, there are still research gaps in applying new data analysis methods (e.g., machine learning) and implementing innovative study designs. In the future, it is desirable to take a holistic approach to precision prevention in the workplace that encompasses all the stages of

**Data Availability Statement:** All relevant data are within the manuscript and its Supporting Information files.

**Funding:** The author(s) received no specific funding for this work.

**Competing interests:** The authors have declared
that no competing interests exist.

precision prevention (risk assessment/data monitoring, data analytics and interventions)
and links them together as a cycle.

## Introduction

According to the World Health Organization [1], non-communicable diseases (NCDs) are
responsible for approximately 74% of deaths worldwide and, therefore, the leading cause of ill
health, disability and death. More than 40% of this burden is preventable [2]. NCDs threaten
the health and well-being of populations globally and convey enormous social, medical, and
economic costs [3]. In addition to unhealthy lifestyle behaviors (e.g., physical inactivity,
unhealthy dietary behavior, smoking and alcohol consumption), NCDs are also influenced by
a complex range of interrelated factors. The individual psychological and physiological profile
and the social and environmental determinants of health (social, economic, cultural, and phys-
ical environments in which people live and work) can have an impact on NCDs and vice versa
[4–7]. Due to this complexity, NCDs remain a major concern [8] and global public health
action requires new approaches to address NCDs [9].

Precision health is one such approach and an emerging field that aims to maximize popula-
tion health and well-being while reducing NCDs and minimizing premature disability and
death [3,10]. This can be achieved primarily through the continuous monitoring of key health
data and generating actionable health discoveries, which will then be used to recommend and
optimize personalized interventions [11].

### From precision medicine to precision health

Precision health, derived from precision medicine, is an extension or progeny of this approach
[12,13]. Precision medicine proposes customizing medical decisions and tailoring treatment to
each individual's unique genetic or biological composition [14–16]. Despite the relevance and
achievements of precision medicine in the treatment of diseases, this approach has some limi-
tations from a health promotion and prevention research perspective. Firstly, precision medi-
cine often reduces the human being to biological data, thus ignoring a holistic biopsychosocial
understanding of health and its numerous determinants [17]. Secondly, precision medicine is
disease-orientated and predominantly concerned with the consideration of sick populations in
the sense of curation (treatment of diseases), thereby lacking the application in health promo-
tion and primary prevention [18].

Although the understanding of precision medicine has expanded in recent years with new
concepts emerging, e.g., exposome science [19,20] or personomics research [21], there is still
the limitation that it is almost exclusively used in the field of secondary or tertiary prevention
[18]. Thus, disease treatment is given priority over prevention [9]. This appears paradoxical
because research has shown that every dollar invested in prevention saves approximately 27
dollars in the long run [22]. For this reason, Hekler et al. [18] noted that the full potential of
precision efforts will only be achieved if precision is applied across the spectrum of health,
including population health and public health [23], not just medicine. Based on these limita-
tions in precision medicine, several approaches have been developed in precision research in
recent years: precision public health, precision health and precision prevention [10,12,13,24–
27].

### Precision prevention

Precision prevention uses biological, behavioral, socioeconomic and epidemiological data to
develop and implement strategies tailored to, e.g., reducing the incidence and mortality of

NCDs in a specific individual or group of individuals [24]. Although the term precision prevention turned up first in 2014, and there is a growing number of studies, the terminological use is still very variable or unclear [13,24]. In the context of using the compound term precision prevention, the most commonly used terms to describe precision are individualized, personalized, tailored, or stratified [13,24]. Biró et al. [24], for example, argue that individualized, personalized and precision prevention are not separate terms but are embedded in each other. In their understanding, precision prevention encompasses and extends both individualized and personalized prevention. Following this broad understanding of precision prevention, health professionals consider not only a person's characteristics and lifestyle (individualized prevention) but also the personal omics profile, as well as the socioeconomic status or the opportunities offered by psychological and behavioral data of the person (personalized prevention) when making proposals to maintain or improve the individual's quality of life [24]. In the present study, we decided to rely on the terminological description of Biró et al. [24] and therefore, use the term precision prevention and its associated definition. However, several publications use the broader concept and term of precision health [3,10,18].

Precision prevention aims to develop an expanded precision perspective and may be viable within the whole prevention paradigm–primary, secondary and tertiary prevention [24]. Precision prevention claims to not just address symptoms but to directly target genetic, biological, environmental, social and behavioral determinants of health [9,18] and to optimize non-pharmaceutical interventions based on these factors [28,29]. This ensures that the right support is provided to the right individual at the right time [18,30–33]. Accordingly, precision prevention focuses on multiple determinants of health and takes a holistic view of health according to the WHO biopsychosocial model of health [34], while also taking a lifespan perspective of health into account, therefore including interventions across the lifespan [10,18].

## The current state of research in precision prevention

Although precision prevention is still in its infancy, research in this field has developed rapidly and particularly early work has produced promising findings [10]. Gambhir et al. [11] have structured the research approaches to precision prevention into a cyclical model that looks at precision prevention in the key components: (1) risk assessment, personal and environmental monitoring; (2) data analytics and (3) tailored interventions [10].

Based on this cyclical model, a scoping review by Viana et al. [10] presented the current state of international research on precision health in a very general and comprehensive way. At the same time, however, it also highlights numerous desiderata in this research area. About half of the studies focused on developing an intervention, primarily individual digital health promotion tools or public health programs, implemented in rural communities, hospitals, workplaces and many other settings. The three most commonly addressed health conditions were metabolic disorders (16%), cardiovascular disease (13%) and cancer (12%), and only 7% aimed at increasing physical activity or weight reduction [10]. Finally, medical approaches currently seem to dominate the research area, as only 15% of the studies can be assigned to the behavioral and social sciences.

In addition to the comprehensive review by Viana et al. [10], Mauch et al. [3] included 31 studies in precision health interventions targeting modifiable health behaviors in their scoping review. Most interventions targeted physical activity (87%) and/or dietary intake (77%). Intervention content was personalized manually (via human interaction) in more than half of the studies (55%) and automatically (via a digital platform) in 35%. The information used to personalize interventions was primarily behavioral or lifestyle data (65%) and physiologic, biochemical or clinical data (48%) [3].

Although these reviews already provide a comprehensive insight into precision prevention, e.g., Viana et al. [10] conclude that their scoping review can serve as a basis for future syntheses of precision health advances. Future reviews should, therefore, first broaden the search strategy thematically in order to better capture emerging sub-disciplines and then focus specifically on certain stages of individuals' lifespans (e.g., children, adults) and settings (e.g., school, workplace). Regarding certain lifespan stages and from the perspective of public health, especially health promotion and prevention, health-promoting programs are particularly feasible and effective when targeting people in different settings, i.e., in their most important living and working environments [35].

## Worksite health promotion and precision prevention

For adults, together with the family and community setting, the workplace setting is a crucial setting for implementing and carrying out health-promoting programs [36,37], with working adults estimated to spend 30% to 40% of their waking time at work [38,39]. Among other things, the workplace provides an efficient structure to reach large groups for health-promoting interventions and uses natural social networks [40]. For example, occupational health services, which are part of the work organization, can be used with their know-how to introduce effective workplace health promotion interventions [41]. For these reasons, workplaces are an ideal setting for health promotion interventions, as they not only offer access to a large number of people but also offer the opportunity to implement interventions at multiple levels targeting individual, organizational and environmental determinants of health and health behaviors [42].

Despite the relevance of the worksite as an ideal setting for health promotion, precision prevention in the occupational context has not yet been considered comprehensively or has only targeted specific health conditions [43,44]. Viana et al. [10], for example, could only identify three studies (3/225; 1%) in their scoping review that were conducted in the workplace [43–45], while the scoping review by Mauch et al. [3] on precision health in the context of behavior change interventions identified one additional study in the workplace setting [46].

In the workplace setting, Moe-Byrne et al. [47] published a review in which they present studies on a thematically very close understanding of precision prevention (tailored interventions). The authors conducted a systematic review of randomized controlled trials (RCTs) to assess the effect of tailored digital health interventions provided in the workplace aiming to improve physical and mental health, presenteeism and absenteeism of employees. The study outcome was positive for tailored digital interventions regarding presenteeism, sleep, stress levels and physical symptoms related to somatization. However, this was less true for tailored digital interventions addressing depression, anxiety and absenteeism. Although these interventions did not reduce anxiety and depression in the general working population, they did significantly reduce depression and anxiety in employees with higher levels of psychological distress [47]. In addition, Moe-Byrne et al. [47] report that most studies showed no improvement in absenteeism but a faster return to work among long-term sick workers who had received a tailored intervention.

## Research gap and objectives

In summary, despite substantial developments in recent years, research on precision prevention is still in its infancy [10]. There is a fundamental lack of reviews that address approaches and interventions in the context of health promotion and primary prevention, recruiting "healthy" or non-diseased populations [10]. Another research gap presents the fact that, to the knowledge of the authors of the present scoping review, there are currently no studies in

which precision prevention is implemented holistically, i.e., according to the model components of Gambhir et al. [11] and the stages of Viana et al. [10]: 1) risk assessment/data monitoring, 2) data analytics and 3) tailored interventions. In addition, there is currently a lack of research on specific populations in their living and/or working environments. Especially in the adult age group, studies on precision prevention in the workplace have not yet been comprehensively presented. However, this setting is particularly relevant regarding health promotion and prevention [36,40,42].

The main objective of this scoping review is to map the current state of precision prevention research in the workplace setting, specifically to:

1. Summarize precision prevention research study context and characteristics (e.g., year of publication, health condition, sample characteristics).

2. Describe and analyze the precision prevention approach in the stages of risk assessment/ data monitoring, data analytics, and the health promotion interventions implemented by study design and data analysis.

   Based on this mapping, in the discussion of the main objective, we:

3. Identify knowledge gaps in precision prevention in the workplace, highlighting promising research directions that can help shape future directions for research and implementation in worksite health promotion.

## Materials and methods

To address the research objectives outlined above, a scoping review was conducted as previously described in the Joanna Briggs Institute Reviewers' Manual [48] and following the reporting guidance provided by the Preferred Reporting Items for Systematic Reviews and Meta-Analyses Extension for Scoping Reviews (PRISMA-ScR [49]). The eligibility criteria and the search strings of this scoping review were pre-registered on the Open Science Framework (OSF) Registration Website (https://osf.io/3xhng/, July 5, 2023).

### Information sources

Information on precision prevention and workplace health promotion was retrieved on two dates (May 4, 2023 and May 15, 2023) due to the workload of the retrieval process from the following six electronic databases: Web of Science$^{TM}$, Scopus$^®$, Embase$^®$, Ovid MEDLINE$^®$, PubMed$^®$ and APA PsycInfo$^®$. Google Scholar$^{TM}$ was not considered since this database did not support the comprehensive search string described below. In addition to searching articles in the databases mentioned above, references of eligible articles were separately screened on August 8, 2023 to detect further topic-related articles that had not been identified by the employed search strings thus far. Within this step, a search for follow-up articles of previously published study protocols detected by the search string was carried out if the string-based literature search did not already identify them. Grey literature was not considered as this review focuses on peer-reviewed publications and the current state of research on precision prevention in workplace health promotion.

### Search strategy

Databases were searched for articles on precision prevention in workplace settings published in English between January 2010 and May 2023. For this purpose, a database search string for titles and abstracts was developed, employing two sets of search term groups. One group of

search terms was chosen to focus on "precision" by also using the search terms "personalized", "individualized", "stratified", "tailored" as described by Viana et al. [10], as well as "targeted". A second group of search terms was chosen to address "prevention" with the additional search terms "health promotion", "health", "program" and "intervention". The Boolean operator "OR" combined search terms of each group.

To identify articles in which search terms of the two groups were employed as term combinations such as "personalized health promotion" or "targeted intervention", this specific search aspect was introduced into the search string by suitable database-specific proximity operators if available, e.g., NEAR/x operator (Web of Science[TM]), PRE/x operator (Scopus[®]), adj (Embase[®] and MEDLINE[®]). For the PsychInfo[®] and PubMed[®] databases, adapted strings were individually programmed to retrieve the search term combinations.

The Boolean operator "AND" finally combined all search term pairs related to "precision"/ "prevention" with the workplace-related terms "worksite", "organizational", "occupational", "worker", "employee" or "corporate", also considering different spellings (e.g., "organizational" or "organisational"). The entire set of electronic search strings adapted to the specific requirements of the six electronic databases are listed in S1 Table 1 in S1 File.

## Selection of sources of evidence–eligibility criteria

The following types of journal articles were included: primary empirical research studies (intervention studies, e.g., RCTs and observational studies, e.g., cross-sectional studies), study protocols and conference proceedings. We included different study types to capture prospective and retrospective designs and even planned studies to include and describe the broad research field of precision prevention in occupational health [50]. Reviews and meta-analyses were not included because they typically considered secondary studies and therefore these studies could be included twice, but the reference list of eligible reviews was screened to identify additional research [51]. Furthermore, editorial articles, book chapters, dissertations, abstracts and posters were excluded due to the lack of a structural quality assessment such as peer reviews. If the full text of articles could not be retrieved, these articles were excluded. Articles found by the search strategy were only included, if they reported on adult participants in workplace settings, human tissue samples (e.g., genetic material) or historical datasets (e.g., health records) of workers. Articles unrelated to workplace settings, such as community settings, were excluded. All eligible articles had to contain descriptions related to "precision prevention" or similar in title and/or abstract, introduction, methods or results. A detailed summary of the eligibility criteria, focusing on population, context (i.e., setting), concept and source of evidence, is listed in S1 Table 2 in S1 File.

## Screening

All articles retrieved from the databases were first exported to the reference manager software EndNote[TM] [Alfasoft GmbH, Germany] for the detection and removal of duplicates and subsequently transferred to the Rayyan App (http://rayyan.qcri.org) [Rayyan Systems, Boston, Massachusetts, USA], a free web-based collaboration platform for systematic reviews [52], which was used for the subsequent screening, data extraction, data synthesis and analysis.

According to the eligibility criteria, screening of articles by title and abstract was performed by all authors in six pair combinations of two reviewing authors, with each pair reviewing one-sixth of all articles. This resulted in a screening workload of 50% for each author.

Firstly, judgments on article eligibility were independently made by all reviewers. Secondly, independent screening results of both reviewer pairs were merged and potential disagreements in judgment were discussed in a team meeting. Where consensus could not be reached, the full

texts of the articles were retrieved for additional evaluation of the introduction, methods, and results regarding the eligibility criteria. Articles that only referred to precision prevention or related terms in the discussion alone were excluded following the notion by Viana et al. [10], which claimed that such articles only addressed the research objective superficially. If a pair of reviewers could not reach a consensus, respective articles were discussed by all reviewers to decide whether to include or exclude an article.

Full texts were subsequently retrieved for all articles identified for inclusion into the scoping review to allow detailed evaluation of the introduction, methods and results section regarding the appropriateness of the reviewers' agreed with preliminary judgments based on title and abstract.

## Data extraction

All authors extracted relevant data from eligible articles, with each author reviewing 25% of the relevant studies. Data extraction focused on general characteristics of the study (e.g., health condition or working context), information on the study's aims, design, main analysis, results and implications for future research, as well as qualitative data related to precision prevention, including measured health outcomes, tailored variables and definitions of precision health.

## Data analysis and synthesis

Data analysis and synthesis were initiated in the Rayyan App based on highlighted and commented text sections relevant to our review's aims. Building on this process, comprehensive data analysis and synthesis were performed in a spreadsheet using Excel® software [Microsoft Cooperation, Redmond, Washington, USA], where the relevant articles' extracted data were transferred. The authors also added comments to this spreadsheet and discussed the content to condense the findings in group meetings and reach an agreement on data synthesis. Lastly, data synthesis was conducted by summarizing the findings of our review in text and tables and visualizing the data in charts using Word® and Excel® software [Microsoft Cooperation, Redmond, Washington, USA].

## Results

### Articles screened

Database searches retrieved 3,249 articles after removing duplicates. After title, abstract and full-text screening, and detection of additional 28 articles in a snowballing approach using the reference lists and citation tracking, 129 included articles (see S2 File for a reference list) remained for data analysis (Fig 1). The data and the categorizations for analyses can be found in S3 File.

### Study context and characteristics

To address the main objective of this study, we summarized the precision prevention research study context and characteristics. The number of articles with precision prevention focus in the work context ascended with a mean of 9 articles per year. Approximately half of the studies ($n$ = 71 of 129; 55%) were published in the years 2018 to 2023, whereas only 20% of the studies ($n$ = 26 of 129) were published in the first five years (2010 to 2014). Most articles ($n$ = 19) were published in 2022 (Fig 2).

With 30 articles, the majority of articles was published by authors from the United States of America (USA with 23%), followed by the Netherlands (11%) and Canada (9%). Fig 3 shows the global distribution of the articles.

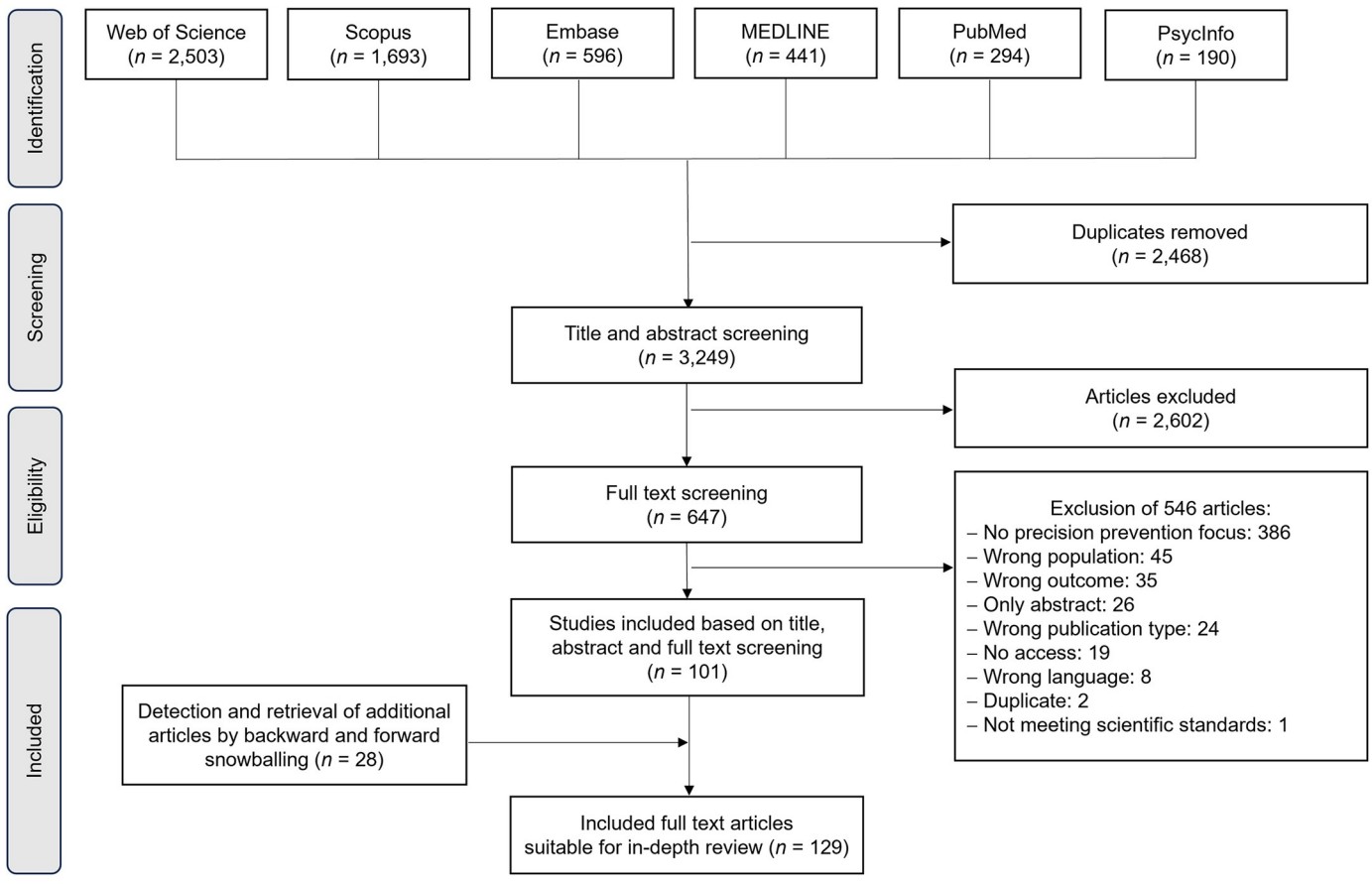

**Fig 1. PRISMA flow chart for the procedure of this scoping review.**

Sample sizes at the baseline of the studies in our included articles varied between two and 12,696 participants, with a median of 169 participants. A priori estimated sample sizes from study protocols were not included. The mean age of participants was 42.5 years (*Median* = 43.6; *Min* = 24.0; *Max* = 57.1 years). For articles that mentioned participant gender (*n* = 94 of 129; 73%), the median percentage of female gender participants was 61%. Six articles only focused on the female gender, while 12 articles focused on the male gender. In 33% of the articles, information about the social background of the participants was given. Most of the articles concentrated on educational level (19%) or occupational level (5%) and only six articles included factors such as income or salary, followed by three articles including ethnicity. Regarding occupational groups (see Table 1), most articles (30%) addressed healthcare work-ers, e.g., hospital and nursing staff. The interventions rarely addressed people with physically demanding jobs, such as working in the agriculture or transport sector (e.g., professional drivers).

## Analysis of the targeted health conditions

Multiple health conditions were mentioned in the analyzed articles (see Table 2). Therefore, 160 health conditions occurred, including health outcomes with a psychological health focus (e.g., mental health, stress, strain or depression) and a physiological health focus (e.g., muscu-loskeletal complaints or obesity). Furthermore, the health conditions could be classified as behavioral outcomes, for example, physical activity/sitting behavior, eating habits, smoking/

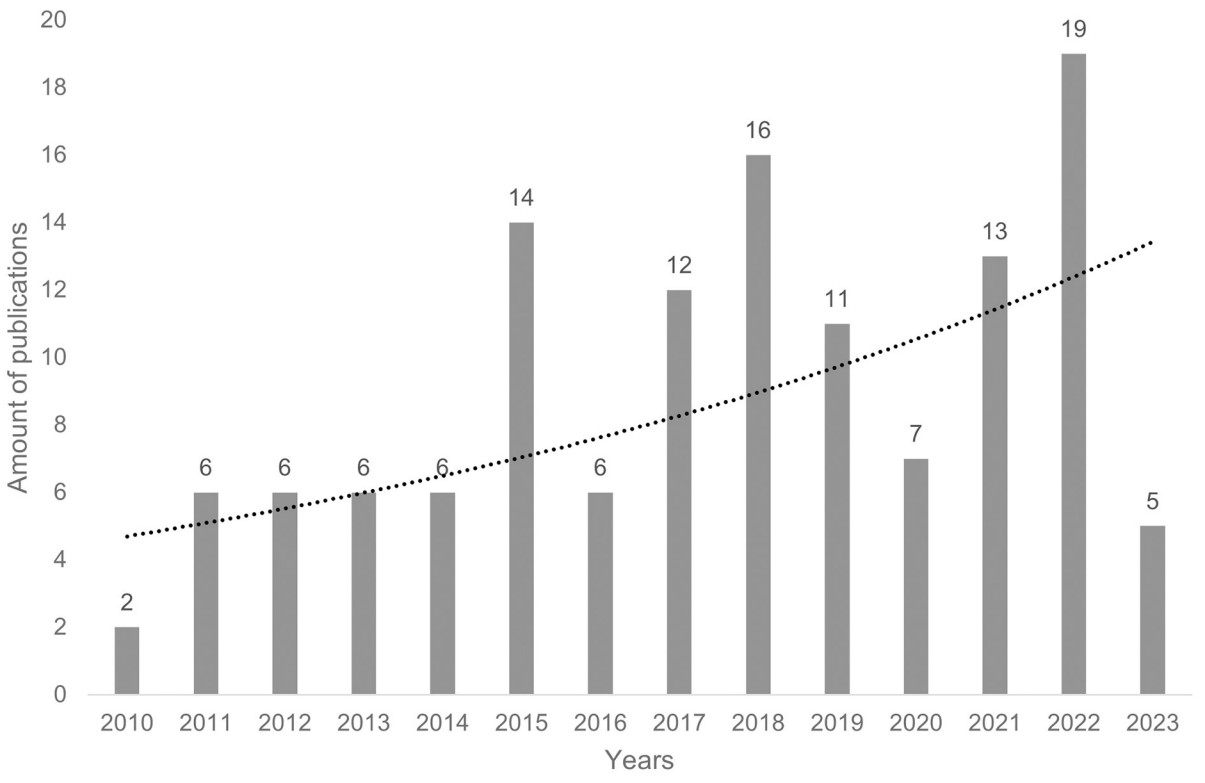

**Fig 2. Published articles per year.**

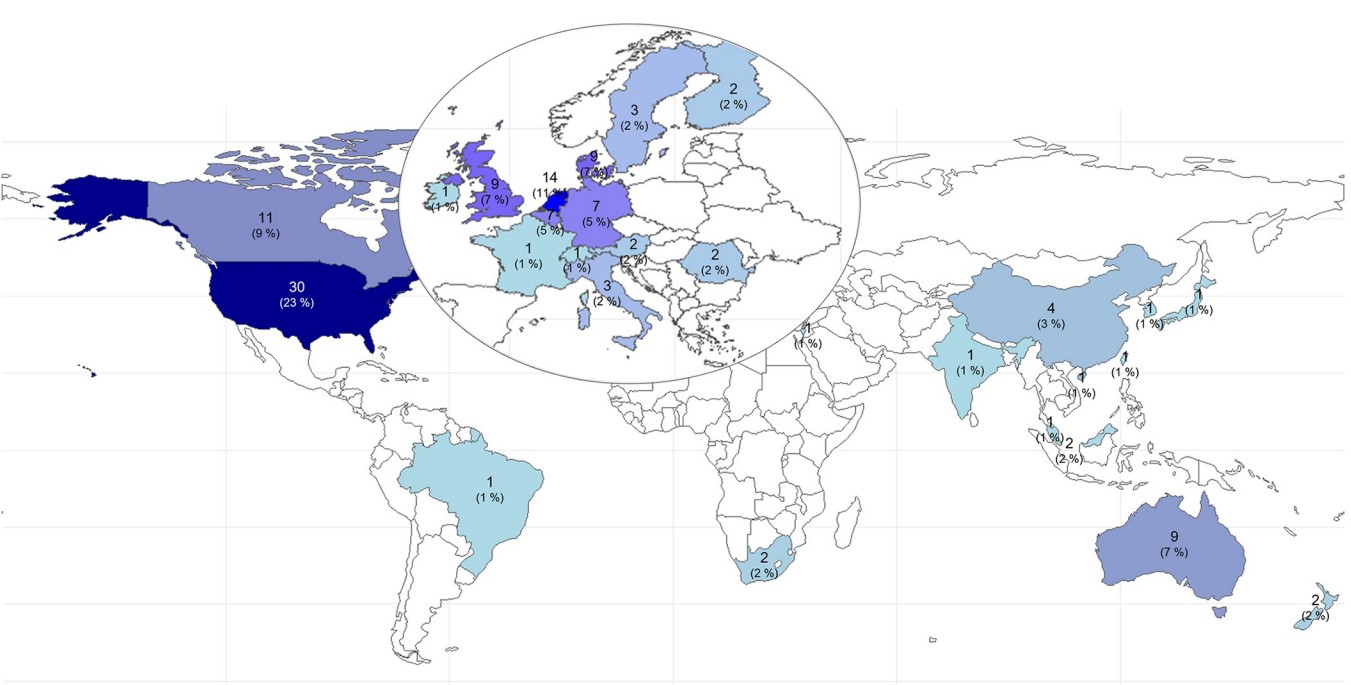

**Fig 3. Amount and (percentage) of publications per country.** The country of origin of the first author was used. A zoom was placed on Europe, as smaller countries in terms of area were difficult to represent; *n* = 129 studies. Made with Natural Earth.

Table 1.  Absolute and relative frequencies of occupational groups (*n* = 162, multiple mentioning possible).

| Health-care | Public | Office | Cons-truction | Service | Trans-portation | Farming/Food | Other |
|---|---|---|---|---|---|---|---|
| 48 | 28 | 23 | 17 | 15 | 7 | 7 | 17 |
| 30% | 17% | 14% | 11% | 9% | 4% | 4% | 11% |

alcohol consumption or work-related outcomes, for example, work performance, job satisfaction and sick leave days of absence as well as communicable diseases such as the Coronavirus disease 2019 (COVID-19).

## Analyzing the stages of the precision prevention approach

Furthermore, we used the model of Gambhir et al. [11] to determine the precision prevention stages and combined the stages 1) risk assessment and data monitoring, followed by 2) data analytics and 3) tailored interventions. In our scoping review with 129 articles, 18 articles (14%) primarily focused on risk assessment/data monitoring, 16 articles (12%) mainly included data analytics studies, and 95 articles (74%) implemented and/or evaluated an intervention. When an article was categorized as risk assessment/data monitoring, we analyzed whether resources, risks or a combination of the two were addressed. Almost half of the articles on risk assessment/data monitoring focused on risks, while one-third of the articles used a combination of risks and resources. In the stage of data analytics, we analyzed the methods of the studies and most of the articles examined a profile approach with latent profile analysis or cluster analysis, followed by machine learning methods.

In the stage of interventions, first, we analyzed how the feedback for tailoring was given, in a way whether the feedback was personal, digital or in a blended manner. Second, we counted how often the feedback was given in interventions, for example, just once or repeatedly (see Table 3). Most interventions were delivered digitally, often in the form of health portals, platforms or tools on websites or in apps. However, the digital interventions also included, for example, a virtual reality-based occupational stress management component [53], automated stress tracking methods [54] or digital behavior change interventions with calendars [55]. If a personal component involving contact with people was used, this usually consisted of consultations and individual counseling. A blended intervention combined these aspects of human contact with digital feedback, either in the form of online consultations on outcomes or an additional opportunity for in-person contact. This is often associated with several feedback loops. For example, the outcome is presented digitally and then discussed in person, whereby the further intervention can be digital again. One of the success factors in reaching employees was the direct use of the intervention in the workplace setting, for example, the delivery of a smoking cessation intervention at a lunch truck [56].

Regarding study designs, more than a third of all studies used randomized controlled trials (39%), followed by cross-sectional studies (18%) and study protocols or pilot studies (12%). Fig 4 shows the distribution of study designs for the different precision prevention stages.

Table 2.  Absolute and relative frequencies of health conditions (*n* = 160 conditions, multiple mentioning possible).

| Behavioral outcomes | Psychological health | Physiological health | Work-related outcomes | Communicable diseases |
|---|---|---|---|---|
| 61 | 37 | 31 | 27 | 4 |
| 38% | 23% | 19% | 17% | 3% |

**Table 3. Categorization of different article types and feedback or tailoring for interventions.**

| Assessment/Monitoring $n = 18$ (14%) | | | Analytics $n = 16$ (12%) | | Intervention $n = 95$ (74%) | | | | | |
|---|---|---|---|---|---|---|---|---|---|---|
| Focus | | | Methods | | Feedback | | | | Tailoring | |
| Resources | Risks | Both | Profiling | ML | Digital | Personal | Blended | | Single | Multiple |
| 4 | 8 | 6 | 11 | 5 | 42 | 30 | 23 | | 74 | 21 |
| 22% | 45% | 33% | 69% | 31% | 44% | 34% | 25% | | 78% | 22% |

ML = Machine learning.

The main data analyses of all studies were regression analyses (44% with analyses of variance or linear mixed models), followed by univariate analyses (Chi-squared tests, Student's *t*-tests, Fisher's test, nonparametric tests; 12%), and thematic analyses (12%). Machine learning methods (Algorithms, Markov Models) were used in 8% of the articles. A profile approach was conducted by latent profile analyses (7%) or cluster analyses (4%). Only 3% of the articles used other analyses like Generalized Estimating Equations or *N*-of-1 trials, whereas 9% of the articles stated no precise main data analysis (N/A). Fig 5 shows the distribution of main data analyses for the different precision prevention stages.

## Discussion

The main objective of this scoping review was to systematically map the current state of precision prevention research in the occupational setting. This includes, on the one hand, summarizing the context and study characteristics of precision prevention research. On the other hand, it involves describing and analyzing the precision prevention approach in the stages of risk assessment/monitoring, data analytics and the health promotion interventions implemented. Based on this, a further objective was to identify current trends and gaps in precision prevention research in worksite health promotion.

### General aspects

More generally, our scoping review shows that contrary to the findings of Viana et al. [10], many studies exist in the broad context of precision prevention research in occupational settings. While Viana et al. [10] found only three studies on precision prevention in the workplace in their scoping review, we identified 129 publications by use of a more specific search terminology. Although there are numerous studies on precision prevention in an occupational setting, there still does not appear to be a unified understanding in all these publications, due to the numerous, sometimes very different definitional descriptions of precision prevention. Nevertheless, this scoping review provides a good overview of the studies published to date on precision prevention, especially in the stages of risk assessment/monitoring, data analytics and interventions [11].

### Study context and characteristics of included studies

The first part of this scoping review's main objective was to provide a systematic summary of precision prevention research in terms of study contexts and characteristics.

**Years of publication.** The number of studies published annually shows a continuous increase since 2010, with smaller spikes just before the COVID-19 pandemic and then again in 2022. The constant rise illustrates an increasing establishment of precision prevention and related terms, which follows the growing research in precision medicine with a specific time lag

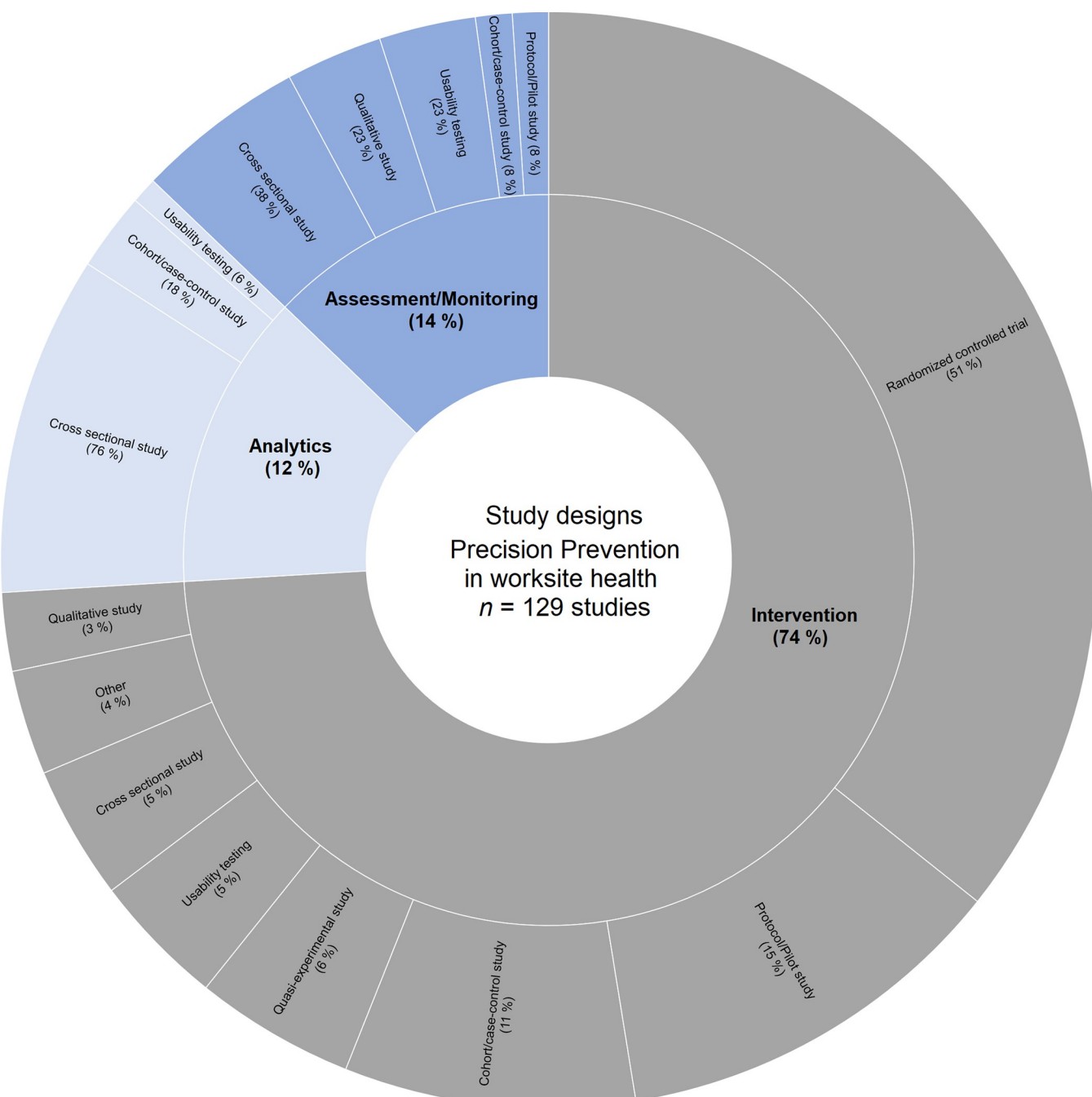

**Fig 4. Overview of studies with precision prevention stages and study designs.** The inner layer displays the stages of precision prevention addressed by the articles and the outer layer represents study designs within each stage of precision prevention, adding up to 100% of the respective stage; (N/A = not available).

[57]. This continuous increase is also consistent with the results of the scoping review by Viana et al. [10]. The authors found that almost half of the identified studies (46%) were published between 2017 and 2020. Although precision prevention has only been established since 2010 [13,10], our results indicate that it has also gained importance in the workplace context since then. Furthermore, the increase in 2022 shows that there has been a rapid application of precision prevention, particularly in the health sector, with a strong focus on COVID-19 disease.

**Countries.** Analogous to the results of Viana et al. [10], the majority of the studies were published in North America (USA and Canada), followed by several European countries (Netherlands, Denmark, United Kingdom) and Australia. Precision prevention seems to have spread mainly in developed economies or high-income countries, while it has yet to establish itself in African and South American countries. The different beginnings of precision prevention research may explain this uneven distribution by country. For example, one of the first

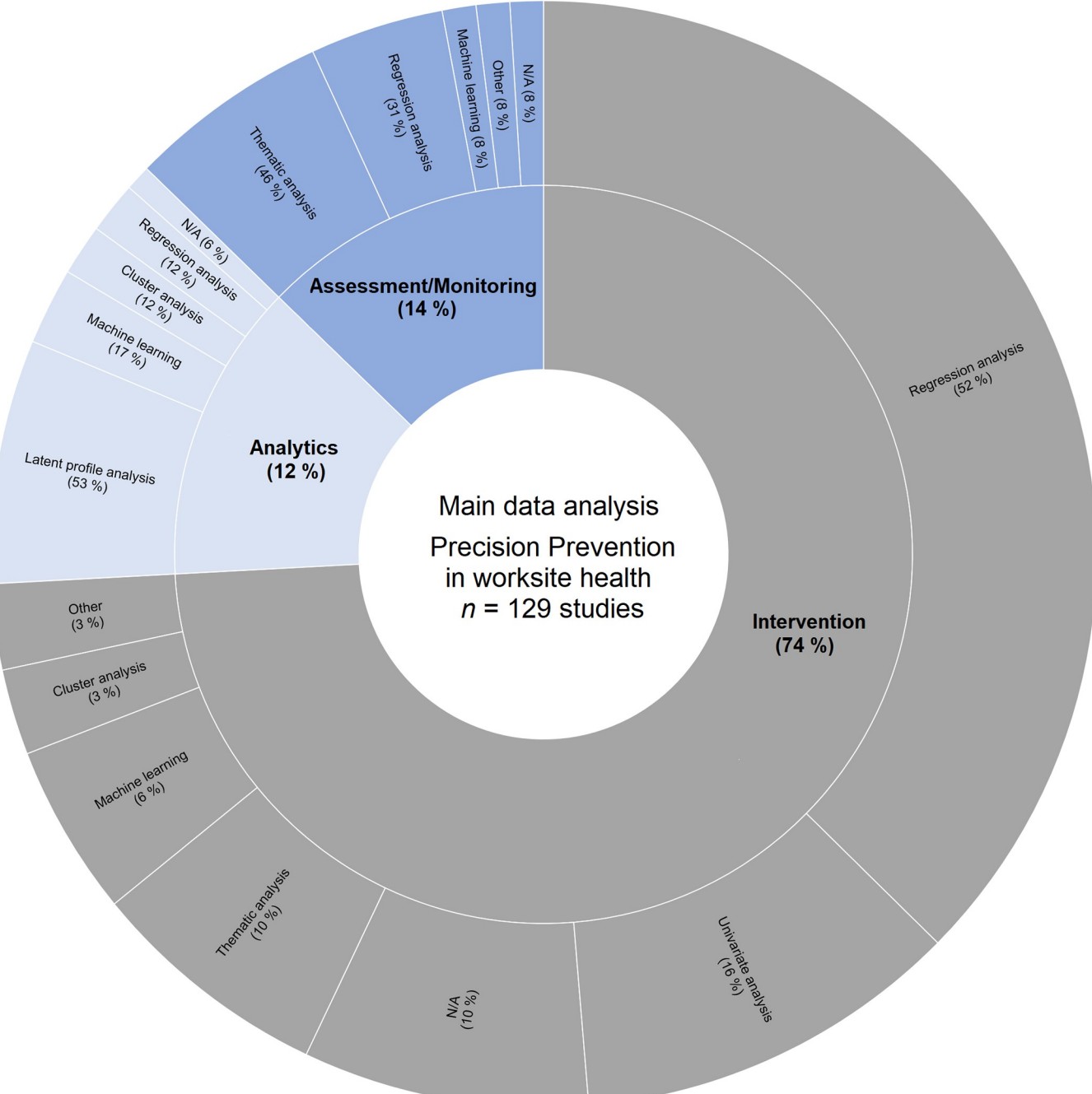

**Fig 5. Overview of studies with precision prevention stages and main data analysis.** The inner layer displays the stages of precision prevention addressed by the articles and the outer layer represents main data analyses within each stage of precision prevention, adding up to 100% of the respective stage; (N/A = not available).

definitions of precision prevention was described by the American Association of Cancer Research in 2016 [24]. Another reason for this early and rapid increase in the North American regions could be that former US President Barack Obama introduced and promoted precision medicine and prevention as a new research focus in 2015 [58]. This initiative fostered the rapid development of precision medicine [27] and, later, precision prevention. Furthermore, this country-specific distribution is consistent with the general research on occupational health management [59–61].

**Participants' characteristics and occupational group.**   Due to the focus on employees, the average age of the participants is 42.5 years, roughly in the middle of working life. This aligns with the results of Viana et al. [10] and other reviews and meta-analyses in occupational settings [62,63]. According to Viana et al. [10], the gender of the participants was only reported in three-quarters (73%) of the studies and full socioeconomic status was rarely reported in the involved studies (2%). However, the participation rate of women in the studies that took gender into account was 61%, so there tends to be an overrepresentation of women here. On the one hand, this can be because many studies were conducted in the health sector, which mainly comprises a female workforce [64]. On the other hand, female employees tend to be more open to addressing health-related issues and participate in workplace health promotion programs more often than men [40,65].

A comparison of the addressed occupational groups shows that most studies were conducted in the health-care sector. Employees with physically demanding jobs (e.g., construction, farming) were observed less frequently. On the one hand, this result may be explained by the comparatively high health needs and occupational requirements of this occupational group, e.g., in nursing [66]. On the other hand, precision prevention has evolved from precision medicine and is therefore strongly related to medical settings, so research is also strongly influenced by the medical health professions [10]. Although construction workers also have high health needs and physical demands compared to other occupational groups [67], our literature review shows few studies on precision prevention in this occupational group. However, this result is consistent with the current literature, as this occupational group is generally rarely addressed in workplace health interventions [68]. The relatively frequent inclusion of employees in the public or office sector can be explained, among other things, by the fact that they tend to have a higher socio-economic status and are therefore more interested in health issues in general and can, therefore, be more easily convinced to participate in health-promoting interventions [69]. This sizeable inclusion of workers with a higher socio-economic status and thus from the public or office sector is consistent with other reviews in the workplace context [69,70].

**Interpretation of the targeted health conditions.**   Most of the studies involved focused on behavioral outcomes, followed by psychological and physiological health outcomes. The intense focus on behavioral outcomes is consistent with research on precision health [10] and research in occupational settings [71]. On the one hand, this strong focus on behavioral outcomes is understandable as it already seems well-established that interventions focusing on behaviors such as dietary behavior, physical activity or smoking are effective in preventing or reducing NCDs [72,73]. On the other hand, it is surprising, as in a recent review of reviews [71], only a small effect of workplace health promotion programs targeting, e.g., physical activity and dietary behavior to reduce workers' body weight was reported.

Work-related outcomes are currently underrepresented in the analyzed studies. This can be explained by the fact that environmental factors, such as the working environment, have so far received little attention in workplace health promotion programs [74]. In addition, it may also be because holistic approaches that combine health-related behavior with environmental changes are rare [70]. Therefore, Robroek et al. [70] call for a better understanding and

research of systems approaches that consider people holistically with their health behaviors and their work and environmental context.

In contrast to the findings of Viana et al. [10], psychological and physiological health outcomes are addressed to roughly the same extent in the studies included in our present review. Accordingly, precision prevention research in the workplace appears to be less coined by medicine, which at times tends to a predominantly biological understanding of health and instead uses a holistic biopsychosocial model of health [18].

Overall, the results of the "study context and characteristics" show that there is currently a need for research into precision prevention both in countries that have been neglected to date (e.g., African and South American countries) and in specific occupational groups (e.g., construction, farming). In addition, behavior-related aspects are currently over-represented in research on precision prevention, while work-related factors have been largely ignored.

## Interpreting the stages of the precision prevention approach

The second part of this scoping review's main objective was to describe and analyze how the precision prevention approach was used in the stages of risk assessment/data monitoring, data analytics and interventions. For this purpose, we used an adapted model from Gambhir et al. [11] to determine the stages in the precision prevention ecosystem on which our identified articles focus. Similar to the review by Viana et al. [10] (47%), most publications in our review applied precision approaches to develop and evaluate interventions (74%). In the stage of data analytics, precision approaches are currently used very little (12%) in the publications identified by our review, but this result is also comparable to Viana et al. [10] (19%). However, in contrast to the results of Viana et al. [10], who found that 35% of the studies target precision prevention approaches in the stage of risk assessment/data monitoring, our review examined 14% of the studies addressing this stage in the worksite context. The high percentage of intervention studies can be explained, among other things, by the fact that some of the studies found are protocol papers and pilot studies, which we assigned to the interventions. The relatively low use of precision approaches in data analytics is surprising, as it forms the "core" between risk assessment/data monitoring and personalized interventions in the model of Gambhir et al. [11]. Without this stage, holistic precision prevention cannot be fully implemented.

**Precision prevention in risk assessment/data monitoring and data analysis.** The results on tailoring in the stage of risk assessment/data monitoring show that almost half of the studies (45%) aim to identify specific risks concerning employees. This intense focus on risks can be explained by the fact that precision prevention research is strongly influenced by medicine and well-established -omics research. Therefore, the theoretical approach is oriented toward disease prevention and less toward health promotion [10]. In the stage of data analytics, machine learning approaches have rarely been used in studies on precision prevention in the workplace context. This relatively low use of machine learning approaches so far can be explained by the fact that they are mostly sensor-based or carried out using certain digital technologies (wearables, webcams, etc.), which can be cost-intensive and entail high data protection requirements [75,76]. Moreover, workers' acceptance of digital technologies depends on many factors and is quite ambivalent [77].

**Precision prevention in interventions.** In the way precision prevention is implemented in occupational health management, it is mostly not the intervention as a whole that is tailored to the employees, but precisely the type of feedback such as personalized recommendations, (digital) messages or tailored feedback ($n = 99$). This feedback is often provided digitally (43%), which is greatly in line with the findings of Mauch et al. [3] and can be attributed to the

fact that digital interventions (wearables, apps, etc.) have become increasingly accepted and utilized in occupational health management in recent years [75,78–81]. Next to the rise of digital interventions, most of the studies were personally delivered, emphasizing the importance of personal contact and interaction for workplace health promotion programs. When planning worksite health promotion programs, the combination and integration of digital and personal components could, therefore, be aimed for in the future [10], as such an interlocking approach between digital and personal feedback would also address concerns about the dehumanization or depersonalization of health through precision health and health information technologies.

The results on the frequency of tailoring in the interventions showed that in most cases (78%), employees were assigned to a specific personalized intervention or were given personal feedback only once, often at the beginning of the intervention. The participants received feedback multiple times or continuous tailoring, especially when digital tools, such as wearables or apps, were used in the interventions. Because the effects of interventions primarily depend on which constructs are tailored or that, e.g., simply sending a message does not necessarily lead to desired changes in employee behavior [82], multiple tailoring should be implemented to a greater extent in the future [18].

**Study designs.** Even though Mauch et al. [3] used a slightly different classification system for 31 intervention studies in their review, the results are comparable in parts. For example, Mauch et al. [3] identified 39% RCTs and 16% cohort studies. One difference in this comparison is that Mauch et al. [3] also included 23% controlled clinical trials (CCT) and 19% longitudinal studies with pre-post-test design, whereas in our review 23% of the articles are descriptive studies (cross-sectional; qualitative). Nevertheless, high-quality study designs, such as RCTs for precision prevention interventions in occupational health management, have not yet been comprehensively established. For example, Moe-Byrne et al. [47] identified only seven RCTs in their review of the effectiveness of tailored digital health interventions for mental health.

The comparison of the study designs with reviews of interventions in occupational health management shows an overall heterogeneous picture. When analyzing the RCT designs as the gold standard for determining intervention effectiveness [83], for example, only 19% of the study designs assessed in the review by Soler et al. [84] were RCTs (individual, group). In contrast, the review by Ni Mhurchu et al. [62] on the effects of workplace health promotion interventions on workers' dietary behavior found 63% of the 16 included studies to be RCTs. In the review by Feltner et al. [85] on the effectiveness of total worker health interventions, 80% of the studies identified were RCTs. Mänttäri et al. [86] also found an almost identical percentage of RCTs (81%) in their review of interventions to promote work ability by increasing physical activity among workers. In comparison, in the review by Anger et al. [87], only 53% of the identified studies were RCTs. Overall, the analysis of the study designs shows that newer designs, such as JITAIs or *N*-of-1 designs [18], are rarely used in occupational health management. If, e.g., the classification of Hekler et al. [18] is applied to interventions that support tailoring, then most interventions are of the *generic* or *tailored* types, whereas *adaptive* or *continuous tuning* interventions are still rare.

**Main data analysis.** Concerning data analytics, established methods such as Chi-squared tests, Student's *t*-tests, and (multivariate) analyses of variance (ANOVA/MANOVA) are currently still predominantly used to identify clusters or phenotypes in the occupational context or to analyze the effects of tailored interventions. Machine learning, algorithms, AI and other new or complex data analytics methods (cluster or latent profile analyses) were rarely used in worksite health promotion programs (11%). One reason for this is that many studies on machine learning, etc., have only been published in the last three years [88], so the latest developments in the field of artificial intelligence (AI), machine learning, etc., have not yet been

able to establish themselves in health promotion in general [89], and thus not in the occupational context either. However, initial studies and reviews, e.g., on the primary prevention of work-related musculoskeletal disorders [88] or sick leave [90], deal with machine learning approaches. Due to the fact that machine learning, AI, etc., show several advantages (e.g., increased availability of online data, low developmental costs) once they become established [89,91], it can be assumed that such new data analysis approaches will be widely used in the upcoming years. It would also be beneficial to adopt some data analysis methods used in phenotyping [92–94], e.g., multichannel mixed membership model (MC3M) using Bayesian inference. Such new data science methods and technologies are needed to promote the application of precision medicine and, thus, precision prevention [93].

In summary, the results on the "stages of the precision prevention approach" show that there are currently mainly publications on interventions in the workplace setting, whereas studies on data analytics, especially using innovative methods such as machine learning or AI, are still rare. In addition, although digital tools have been used several times for interventions in occupational health, they have rarely been combined with innovative study designs (e.g., adaptive or continuous tuning interventions).

## Strength and limitations

The strength of this review lies in the comprehensive search that used a variety of synonyms for precision prevention [24] to capture relevant studies in the occupational setting. Another strength is the consistent, systematic approach and rigor of this scoping review, in which each step was carried out independently and at least in duplicate. The consideration of a two-sided snowballing to identify further relevant studies can be seen as a further methodological strength.

Nevertheless, we cannot exclude the possibility that further relevant studies could not be identified. Thus, despite our comprehensive search term, it is conceivable that the inclusion of other scientific disciplines, such as occupational safety, and therefore, additional terms would identify further studies. Another limitation of this scoping review is that we did not consider a few types of publications (books, dissertations, grey literature, etc.) and that we have focused on English-language publications only. Finally, another limitation is that we performed the screening in pairs, but not the data extraction, which may lead to errors or inaccuracies in the extracted data. Therefore, this scoping review can serve as a basis for further synthesis of precision prevention in the occupational setting, expanding the current review's scope and filling in its limitations. Future reviews should include non-English publications and expand the search strategy to better capture emerging subdisciplines such as *precision public health*, *precision mental health* [10] or economic aspects of precision prevention.

## Implications and future research directions

Based on these findings, a further aim of this scoping review was to identify current trends and knowledge gaps in precision prevention research in the workplace and to suggest future research directions. Future research should initially focus on developing a unified understanding of precision prevention in the occupational setting and on clearly describing and defining this still very broad construct. This could lead to precision prevention becoming more established and applied in worksite health promotion in the future, both in research and in practice. Regarding the objectives of this scoping review, the following four steps would be recommendable in future research:

1. In the future, precision prevention research should be increasingly implemented in regions and sectors that have been neglected so far to better exploit health potentials in the

workplace setting. More widespread use could also lead to better targeting of vulnerable groups, especially workers from low socioeconomic groups, for health-promoting interventions, thereby better promoting their health and ultimately helping reduce health inequalities [70]. In this context, it would also be desirable, if a holistic, integrated understanding of health is considered in precision prevention research, considering different data sources to identify health needs and target groups. For example, determinants of health behaviors, behaviors and health or work-related outcomes of workers should be assessed and analyzed together in the future, as well as their environmental and societal factors [70]. According to the latest findings in personomics and exposome research [19–21], all these factors have important effects on individuals, which is why precision prevention must include information about a person's environment and living conditions in particular, in addition to personal information. Ultimately, linking the well-founded findings of medical -omics research with the behavioral and environmental findings of social and behavioral science research would make a decisive contribution to considering human health in the future not only holistically but also in terms of reciprocal human-environment interaction.

2. Based on such a comprehensive understanding of health and a more holistic database of employees [24], new methods of data analysis could then also be increasingly used. In particular, approaches in machine learning, algorithms, AI and other person-centered methods could be used more extensively. An example of such an innovative approach is using machine learning to identify psychosocial-behavioral phenotypes by Burgermaster and Rodriguez [94]. To achieve this goal, it is also necessary to involve different expertise in science and practice, e.g., health science, sports science, psychology, medicine, etc. Such inter- and transdisciplinary collaboration would reduce the current dominance of medicine and the often associated one-sided view of precision prevention and expand a holistic approach in precision prevention research [10], thereby promoting further potential.

3. Although most of the 129 identified studies tested or evaluated an intervention, more innovative study designs should be used for interventions in the future to meet the demands of a precision prevention approach–that means targeting the right intervention to the right population at the right time [58,95]. For example, adaptive or continuous-tuning interventions [18], such as JITAIs (just-in-time adaptive interventions), MRTs (micro-randomization trials), *N*-of-1 designs/idiographic designs or even system identification experiments have rarely been tested or evaluated in worksite health promotion interventions. As there are already some best practice studies for the implementation of JITAIs/MRTs (*HeartSteps* [96,97]; *SitCoach–office workers;* [98]), or of system identification approaches (*MyBehaviour*, *Just Walk* [99]), with promising effects on health behaviors or even workers' health, such study designs with the use of digital technology (apps, health portals, health platforms, etc.) should be used even more in the future. The use of digital technologies offers numerous potentials for implementing the precision prevention approach, as, for example, employees can be reached regardless of their place of work and digital interventions can also be individually and continuously adapted to the needs of employees.

4. Furthermore, interventions should not only target individuals (e.g., health behaviors) but also address changes in the environment and society, as such system approaches are particularly promising but are currently rarely used in workplace health promotion [70]. Future studies should therefore be guided by the latest findings in personomics and exposome research [19–21] and, for example, include environmental exposures and living conditions of employees in behavioral interventions, and specifically try to change them in interventions.

In addition to these recommendations for research, it would also be important in the future for stakeholders from occupational practice and also from politics (health, economy, labor, etc.) to address the possibilities and opportunities of precision prevention in occupational health and to shape this field in the future.

In the future, stakeholders from occupational practice could, among other things, focus on the numerous possibilities and opportunities of innovative, digital technologies in health-promoting measures and utilize them even more. In recent years, the numerous changes in work (digitalization, work from home, etc.) have meant, among other things, that many employees (especially white-collar workers) work much less frequently at a fixed company location and are therefore hardly accessible for health-promoting programs on site. In this respect, the greater use of digital technologies in health-promoting interventions (apps, wearables, health platforms, messengers, etc.) could, on the one hand, help to ensure that employees can regularly participate in these programs regardless of where they work. On the other hand, these interventions can also be tailored to the individual needs and preferences of employees in terms of precision prevention. In order to be able to utilize the potential of precision prevention in occupational practice even better in the future, it is also important to improve the quality of counseling in the company, especially in occupational medicine [100]. This ensures that employees receive health-related recommendations that they can understand with their individual health literacy and implement in their everyday professional and private lives.

In order to support awareness and greater use of precision prevention approaches in occupational health, it would be desirable, for example, for health policy to promote financial or tax incentives for the use of digital technologies in health-promoting programs. In addition, it would also be important for politicians to initiate large-scale campaigns, such as former US President Obama with the precision medicine initiative, to further strengthen the importance of precision prevention in society.

## Conclusions

By using an extended search term, our review shows that precision prevention in the workplace setting is more widespread than the review by Viana et al. [10] suggests, which identified three publications. However, one main challenge at present is that the concept of precision prevention in the workplace context has not yet been uniformly defined. Nevertheless, this scoping review provides a good overview of the studies published to date on precision prevention in worksite health promotion, particularly in the stages of risk assessment/data monitoring, data analytics and interventions [11]. In summary, in addition to further developing precision prevention in the four points outlined above, it is also crucial in the future to establish a closer exchange between those involved in science and practice. Because only through such close cooperation can it be ensured, that the scientific considerations and further developments in precision prevention can also be implemented and applied in occupational practice. At the same time, however, it will also be important for science in the future to gain an even better understanding of the framework conditions and, for example, barriers in operational practice (time, personnel, policy, etc.) and to incorporate these findings into the planning and implementation of research projects in the field of precision prevention. Such a joint approach between science and practice will not only drive the further development of precision prevention in occupational health but can ultimately also help to ensure that the knowledge gained flows even more strongly into health policy or, for example, into economic and labor policy in the future, and can thus contribute to fundamental changes and positive developments in the context of health and work.

## Supporting information

**S1 File. Search strings and eligibility criteria.**
(PDF)

**S2 File. Reference list of included articles.**
(PDF)

**S3 File. Data for the results.**
(XLSX)

**S4 File. PRISMA-ScR.**
(PDF)

## Acknowledgments

We want to thank Anna Kaufmann for her assistance with revising the manuscript language.

## Author Contributions

**Conceptualization:** Filip Mess, Simon Blaschke, Teresa S. Schick, Julian Friedrich.

**Data curation:** Simon Blaschke, Julian Friedrich.

**Formal analysis:** Filip Mess, Simon Blaschke, Teresa S. Schick, Julian Friedrich.

**Investigation:** Filip Mess, Simon Blaschke, Teresa S. Schick, Julian Friedrich.

**Methodology:** Filip Mess, Simon Blaschke, Teresa S. Schick, Julian Friedrich.

**Supervision:** Filip Mess, Julian Friedrich.

**Visualization:** Simon Blaschke, Teresa S. Schick, Julian Friedrich.

**Writing – original draft:** Filip Mess, Simon Blaschke, Teresa S. Schick, Julian Friedrich.

**Writing – review & editing:** Filip Mess, Simon Blaschke, Teresa S. Schick, Julian Friedrich.

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
