## [Decision Letter · Decision Letter 0]

16 Apr 2024

PONE-D-23-40011Precision Prevention in Worksite Health – A Scoping Review on Research Trends and GapsPLOS ONE

Dear Dr. Friedrich,

Thank you for submitting your manuscript to PLOS ONE. After careful consideration, we feel that it has merit but does not fully meet PLOS ONE’s publication criteria as it currently stands. Therefore, we invite you to submit a revised version of the manuscript that addresses the points raised during the review process.

>

We look forward to receiving your revised manuscript.

Kind regards,

Yashendra Sethi

Academic Editor

PLOS ONE

Journal Requirements:

2. We note that Figure 3 in your submission contain [map/satellite] images which may be copyrighted. All PLOS content is published under the Creative Commons Attribution License (CC BY 4.0), which means that the manuscript, images, and Supporting Information files will be freely available online, and any third party is permitted to access, download, copy, distribute, and use these materials in any way, even commercially, with proper attribution. For these reasons, we cannot publish previously copyrighted maps or satellite images created using proprietary data, such as Google software (Google Maps, Street View, and Earth). For more information, see our copyright guidelines: http://journals.plos.org/plosone/s/licenses-and-copyright.

a. You may seek permission from the original copyright holder of Figure 3 to publish the content specifically under the CC BY 4.0 license.  

Additional Editor Comments:

Dear authors,

I know the review process has taken a long time, but we have now received reports from two subject experts. They have raised some important concerns. Please address them before we can proceed further.

Reviewers' comments:

Reviewer's Responses to Questions

**Comments to the Author**

1. Is the manuscript technically sound, and do the data support the conclusions?

Reviewer #1: Yes

Reviewer #2: Partly

2. Has the statistical analysis been performed appropriately and rigorously? 

Reviewer #1: Yes

Reviewer #2: N/A

3. Have the authors made all data underlying the findings in their manuscript fully available?

Reviewer #1: Yes

Reviewer #2: Yes

4. Is the manuscript presented in an intelligible fashion and written in standard English?

Reviewer #1: Yes

Reviewer #2: Yes

5. Review Comments to the Author

Reviewer #1: This is an excellent paper and it will be very helpful for future researchers and practitioners working on occupational health or precision healthcare. Despite many strengths this paper has, there are a few areas that require clarity as well as further explanations as follows:

1. In the methods, please explain which study designs were considered eligible for inclusion. I see that you included articles by their publication types (e.g., original paper, conference proceedings, but not reviews). Future readers may wish to know if you wanted to include intervention studies or observational ones, or both. And more importantly, why such a choice was made? Please explain those in the methods. In addition, scoping reviews are generally very inclusive and they often include diverse papers, including reviews and editorials. Please explain why you did not include them, and considered protocols even if they did not have any results yet.

2. The results of the paper is very organized, and perhaps one of the best I have seen in scoping reviews. One suggestion I'd make is reflecting on the results more critically, providing comparisons and critical gaps between groups/subgroups of studies in your discussion. For instance, each sub-section of your discussion can/should provide specific research progress and gaps (which you can link with your recommendations later).

3. The idea of precision health, overall, has deep roots in recent progress in "omics"- which you mentioned in your paper as well. However, your paper may provide some reflections (in the discussion section), outlining/summarizing/criticizing the current states of "omics" deployed in existing research. Also, the growth of digitalization is transforming personalized healthcare, therefore, please discuss how such technologies are shaping precision worksite preventions, and provide specific recommendations for enhancing such applications.

4. Your recommendations are well structured- please consider adding a section for implications for policy and practice. Scoping reviews provide mapping of research that promotes further research, and perhaps more notably, inform decision-makers to leverage the available research evidence. By providing insights/recommendations for practitioners, you'll show the real-world value of the mapped evidence that can inform evidence-based practice.

Thanks again for conceptualizing and completing this beautiful review, and I look forward to seeing how this work helps scholars and communities in the future.

Reviewer #2: Thank you for the opportunity to participate in this review. The manuscript features a scoping review focused on precision prevention research in the worksite setting. I hope that the comments that follow will aid the authors in improving the potential contribution of this manuscript.

General

•While the introduction is helpful to set up the stage for the review, I wonder if authors could summarise and trim the text to streamline the key messages, particularly for the sections ‘From precision medicine to precision health’ and ‘Precision prevention’. It seems the same ideas are repeated throughout – I feel they could be combined into one section.

•Methods are well detailed and authors applied an exhaustive scoping review approach. I just lacked a few details (see the ‘specific comments’ section below).

•For Results it would be good to see a more thorough description of the interventions that were implemented. It is surprising that considering the majority of studies identified were interventions, there is so little information about them (e.g., what specific intervention techniques were employed?). Different types of feedback are mentioned, but to me this is just one of many techniques that an intervention could apply (see, for example, the 281 techniques identified as part of the Behaviour Change Techniques Ontology; Marques et al., 2023). How about the rest of techniques? How about the duration of the interventions, their mode of delivery, etc.

•Discussion: this is a long discussion section which could be also trimmed to streamline the key messages. As an example, the section on ‘study designs’ is going back and forth, presenting a multitude of review findings while it is not clear what is the point that authors want to make.

•I cannot really see the figures as they are blurred, and I cannot download the file so I have not had the chance to review them.

Specific

•Lines 26-27: “Over three quarters of the studies addressed an intervention (95/129, 74 %).” 74% is not over three quarters.

•Lines 31-32: “while multiple mentioning was possible”. Not sure ‘while’ is the best connector here. I think authors just want to say something like ‘please note’ here.

•Lines 36-37: “(Algorithms, Markov Models)” for clarity please add ‘e.g.,’ if these are examples.

•Lines 38-39: “Although there is a growing number of precision prevention studies…”. Some of the conclusions presented here do not follow from the results section. For example, the growing number of studies. Please ensure this section maps 1:1 onto the study findings in ‘results’.

•Lines 48-49: “According to the World Health Organization [1], globally, non-communicable diseases (NCDs) are responsible for approximately 74 % of deaths worldwide…”. Having ‘globally’ and ‘worldwide’ in this sentence seems redundant.

•Lines 48-49: “Also, due to this complexity, the prevalence of NCDs and their associated impact on public health continue to be a major concern [8] and global public health action requires new approaches to address NCDs [9].”. The first part of this sentence repeats info already presented in the same paragraph (at the beginning).

•Line 63: “This will be achieved primarily…”. Present tense would be more indicated here.

•Line 85: “targeting health behaviors (e.g.,nutrition, physical activity, smoking)”: Nutrition is not a behaviour.

•Line 88: “will be achieved only if”, please rephrase as “will only be achieved…”.

•Line 139: “a scoping review by Viana et al. [10] presents…”. Please use past sentence to introduce findings from previous studies.

•Lines 160-161: “Therefore, future reviews should broaden the search strategy to better capture emerging sub-disciplines and focus on specific stages of individuals’ lifespans.” This seems contradictory. Should they broaden the scope or should they actually do the opposite and narrow it down to capture specific stages?

•Lines 186-187: In contrast to this lack of studies on precision prevention in worksites, Moe-Byrne et al…”. I do not understand what the point is in presenting the findings for this review focused on RCTs. How this contributes to the review and/or the overall rationale?

•Line 229: “To address the issues outlined above…” I suggest a better word-choice here. ‘Issue’ has a negative connotation (i.e., a problem).

•Line 233: “The scoping review was pre-registered on the Open Science Framework”. Authors could be more specific here and state that the eligibility criteria and the search strings were pre-registered. These are the only things I could find in the OSF page.

•Line 245: “Furthermore, a search for follow-up articles of previously published study protocols detected by the search string…”. Does this explain the two searches (May 4 and 15)? Please clarify.

•Line 252: “in English between January 2010 and May 2023”. Is there any rationale for setting up the start of the search as of 2010? Please explain.

•Line 252: “A detailed summary of the eligibility criteria, focusing…”. Could this table be included in the main text? it seems quite important to understand the review.

•Line 384: “Furthermore, we used the model of Gambhir et al. [11] to determine…”. Authors could briefly explain these different stages and include this in Methods, considering the important use of this framework to classify research in the scoping review.

•Lines 430-433: “Although there seem to be numerous studies on precision prevention in an occupational setting…” I don’t understand what is the authors’ point here. Please rephrase.

•Lines 445: “Years of publication”. I understand this is a sub-heading of ‘Study context and…’? If so it would be great to have different heading levels for clarity.

•Line 653: “Strength and limitations”. Authors should include here that, unlike screening, data extraction was not done by pairs, with the consequent risk of errors / inaccuracies in the extracted data.

•Line 731: “identifying three identified publications.” Requires rephrasing.

•Lines 736-740: “In summary, in addition to further developing precision prevention in the four points…”. These last two sentences are not clear and to me do not follow from anything that has been said in the discussion. A stronger conclusion could be to focus on the recommendations for future research.

References

Marques, M. M., Wright, A. J., Corker, E., Johnston, M., West, R., Hastings, J., ... & Michie, S. (2023). The behaviour change technique ontology: Transforming the behaviour change technique taxonomy v1. Wellcome open research, 8.

6. PLOS authors have the option to publish the peer review history of their article (what does this mean?). If published, this will include your full peer review and any attached files.

Reviewer #1: **Yes: **M. Mahbub Hossain

Reviewer #2: No

---

## [Author Response · Author response to Decision Letter 0]

3 May 2024

Dear Dr. Sethi,

Dear Reviewers,

Thank you for the opportunity to improve our manuscript. We appreciate the reviewers’ positive feedback on the design of our scoping review and the presentation of the results. We thoroughly answered the additional requirements of the journal and the reviewers’ questions. We uploaded a marked-up copy of our manuscript that tracks the changes and a clean version with yellow highlighting of the revised manuscript.

We look forward to the further process.

Sincerely,

Prof. Filip Mess, Simon Blaschke, Dr. Julian Friedrich

Journal Requirements:

Response: Thank you for the style requirements that we checked again. Furthermore, we used the Preflight Analysis and Conversion Engine to check the requirements of the figures.

2. We note that Figure 3 in your submission contain [map/satellite] images which may be copyrighted. All PLOS content is published under the Creative Commons Attribution License (CC BY 4.0), which means that the manuscript, images, and Supporting Information files will be freely available online, and any third party is permitted to access, download, copy, distribute, and use these materials in any way, even commercially, with proper attribution. For these reasons, we cannot publish previously copyrighted maps or satellite images created using proprietary data, such as Google software (Google Maps, Street View, and Earth). For more information, see our copyright guidelines: http://journals.plos.org/plosone/s/licenses-and-copyright.

a. You may seek permission from the original copyright holder of Figure 3 to publish the content specifically under the CC BY 4.0 license. 

Response: Thank you for this notification and the helpful suggestions for replacing the copyrighted map figure. We replaced the figure using the R package “natural earth,” which uses “Natural Earth,” as you suggested. “Natural Earth” is a public domain map dataset including vector country and other administrative boundaries. According to the terms of use (https://www.naturalearthdata.com/about/terms-of-use/) no permission is needed to use Natural Earth. Crediting the authors is unnecessary. Nevertheless, we added the short text in the Figure 3 captions: “Made with Natural Earth.”

 

Responses to Reviewer #1: 

This is an excellent paper and it will be very helpful for future researchers and practitioners working on occupational health or precision healthcare. Despite many strengths this paper has, there are a few areas that require clarity as well as further explanations as follows:

Response: Dear Reviewer, 

Thank you very much for your very detailed and appreciative feedback. We very much appreciate the fact that you have taken so much time to analyze our paper in detail and make suggestions that will improve the quality of our publication.

1. In the methods, please explain which study designs were considered eligible for inclusion. I see that you included articles by their publication types (e.g., original paper, conference proceedings, but not reviews). Future readers may wish to know if you wanted to include intervention studies or observational ones, or both. And more importantly, why such a choice was made? Please explain those in the methods. In addition, scoping reviews are generally very inclusive and they often include diverse papers, including reviews and editorials. Please explain why you did not include them, and considered protocols even if they did not have any results yet.

Response: Thank you for this valuable remark. We clarified in the methods (selection of sources of evidence – eligibility criteria) the inclusion and exclusion of the different study types (ll. 268-277): “The following types of journal articles were included: primary empirical research studies (intervention studies, e.g., RCTs and observational studies, e.g., cross-sectional studies), study protocols and conference proceedings. We included different study types to capture prospective and retrospective designs and even planned studies to include and describe the broad research field of precision prevention in occupational health [50]. Reviews and meta-analyses were not included because they typically considered secondary studies and therefore these studies could be included twice, but the reference list of eligible reviews was screened to identify additional research [51]. Furthermore, editorial articles, book chapters, dissertations, abstracts and posters were excluded due to the lack of a structural quality assessment such as peer reviews.”

2. The results of the paper is very organized, and perhaps one of the best I have seen in scoping reviews. One suggestion I'd make is reflecting on the results more critically, providing comparisons and critical gaps between groups/subgroups of studies in your discussion. For instance, each sub-section of your discussion can/should provide specific research progress and gaps (which you can link with your recommendations later).

Response: Thank you very much for this very helpful feedback. We discussed this point intensively internally, as the second reviewer suggested the opposite and recommended shortening the discussion. In order to take both pieces of feedback into account and improve the quality of the paper, we have included summarizing aspects in the discussion section at the end of each of the two sections, which in turn are linked to the recommendations (e.g. ll. 656-661). At the same time, we have shortened and sharpened the discussion in some points.

3. The idea of precision health, overall, has deep roots in recent progress in "omics"- which you mentioned in your paper as well. However, your paper may provide some reflections (in the discussion section), outlining/summarizing/criticizing the current states of "omics" deployed in existing research. Also, the growth of digitalization is transforming personalized healthcare, therefore, please discuss how such technologies are shaping precision worksite preventions, and provide specific recommendations for enhancing such applications.

Response: Thank you very much for these two important responses and additions. We mentioned -omics research in the introduction and deliberately decided against a detailed explanation because, as you write, it has become very important in the medical field in recent years. However, as Precision Prevention focuses strongly on social and behavioral aspects, we have placed these in the foreground. Nevertheless, we were very happy to take up your suggestion and include the references to -omics research in the discussion (e.g. l. 572, ll. 708-712). After intensive internal discussion, we came to the conclusion that we should not add the findings on -omics research comprehensively and in several places, as this would otherwise dilute the focus of the paper and the stringency could be lost. We hope that this solution is appropriate.

We were also very happy to implement your suggestion to take a closer look at digitalization. As we had already included detailed aspects of digitalization in the discussion, particularly in the section "Precision prevention in interventions," we have now added further aspects to the implications (ll. 592-597).

4. Your recommendations are well structured- please consider adding a section for implications for policy and practice. Scoping reviews provide mapping of research that promotes further research, and perhaps more notably, inform decision-makers to leverage the available research evidence. By providing insights/recommendations for practitioners, you'll show the real-world value of the mapped evidence that can inform evidence-based practice.

Response: Thank you very much for this very helpful feedback. We were very happy to take up this suggestion and have added a longer section to the implications, which includes recommendations for both practice and policy. We have also included further additions in the conclusions (e.g. ll. 785-798).

Thanks again for conceptualizing and completing this beautiful review, and I look forward to seeing how this work helps scholars and communities in the future.

 

Response to Reviewer #2: 

Thank you for the opportunity to participate in this review. The manuscript features a scoping review focused on precision prevention research in the worksite setting. I hope that the comments that follow will aid the authors in improving the potential contribution of this manuscript.

Response: Dear Reviewer,

Thank you very much for your very detailed and appreciative feedback. We very much appreciate the fact that you have taken so much time to analyze our paper in detail and make suggestions that will improve the quality of our publication.

General

• While the introduction is helpful to set up the stage for the review, I wonder if authors could summarise and trim the text to streamline the key messages, particularly for the sections ‘From precision medicine to precision health’ and ‘Precision prevention’. It seems the same ideas are repeated throughout – I feel they could be combined into one section.

Response: Thank you very much for your feedback, which we have gladly implemented. First of all, we have modified and sharpened the introduction in several sections, especially in the two sections "from precision medicine to precision health" and "precision prevention". However, we have retained the structure with the subheadings, as we believe that such an outline provides the reader with a better overview. This structure also seems important to us because the "Precision prevention" section in particular contains further thematically appropriate subsections.

• Methods are well detailed and authors applied an exhaustive scoping review approach. I just lacked a few details (see the ‘specific comments’ section below).

• For Results it would be good to see a more thorough description of the interventions that were implemented. It is surprising that considering the majority of studies identified were interventions, there is so little information about them (e.g., what specific intervention techniques were employed?). Different types of feedback are mentioned, but to me this is just one of many techniques that an intervention could apply (see, for example, the 281 techniques identified as part of the Behaviour Change Techniques Ontology; Marques et al., 2023). How about the rest of techniques? How about the duration of the interventions, their mode of delivery, etc.

Response: Thank you for your feedback. Due to the nature of a scoping review, we presented the research field on precision prevention more broadly and not very specifically on interventions, for example, as in a systematic review. Therefore, although the interventions are mentioned as the most identified studies, they are broken down less systematically. We have used the mode of delivery in the form of digital/personal/blended feedback, as this is an aspect that can be clearly identified and provides unambiguous findings. Due to their breadth, all of the interventions cannot be directly assigned to a common categorization, for example, behavior change techniques and their subcategories. Due to your valuable feedback, we have included examples of the respective modes of delivery in the text (ll. 400-412): “Most interventions were delivered digitally, often in the form of health portals, platforms or tools on websites or in apps. However, the digital interventions also included, for example, a virtual reality-based occupational stress management component [53], automated stress tracking methods [54] or digital behavior change interventions with calendars [55]. If a personal component involving contact with people was used, this usually consisted of consultations and individual counseling. A blended intervention combined these aspects of human contact with digital feedback, either in the form of online consultations on outcomes or an additional opportunity for in-person contact. This is often associated with several feedback loops. For example, the outcome is presented digitally and then discussed in person, whereby the further intervention can be digital again. One of the success factors in reaching employees was the direct use of the intervention in the workplace setting, for example, the delivery of a smoking cessation intervention at a lunch truck [56].”

• Discussion: this is a long discussion section which could be also trimmed to streamline the key messages. As an example, the section on ‘study designs’ is going back and forth, presenting a multitude of review findings while it is not clear what is the point that authors want to make.

Response: Thank you for this very helpful feedback. We discussed this point intensively internally, as the second reviewer suggested the opposite and recommended expanding the discussion in some places and, for example, presenting future research aspects in more detail. In order to take both types of feedback into account and improve the quality of the paper, we have shortened the discussion in some places and thus sharpened it. In addition, we have included some summarizing aspects in the discussion section at the end of the two overarching sections (e.g. ll. 592-597).

• Specific

Response: Thank you very much for this very detailed and extremely helpful feedback and suggestions. We have implemented these in the manuscript and also commented briefly on some of them below. With the tracking function, 

---

## [Editor Report · Decision Letter 1]

22 May 2024

Precision Prevention in Worksite Health – A Scoping Review on Research Trends and Gaps

PONE-D-23-40011R1

Dear Dr. Friedrich,

We’re pleased to inform you that your manuscript has been judged scientifically suitable for publication and will be formally accepted for publication once it meets all outstanding technical requirements.

Kind regards,

Yashendra Sethi

Academic Editor

PLOS ONE

Additional Editor Comments (optional):

Thank you for making all the edits - i can see most reviewer comments have been addressed well and hence happy to accept it. Congratulations on your contribution.
---

## [Editor Report · Acceptance letter]

30 May 2024

PONE-D-23-40011R1 

PLOS ONE

Dear Dr. Friedrich, 

I'm pleased to inform you that your manuscript has been deemed suitable for publication in PLOS ONE. Congratulations! Your manuscript is now being handed over to our production team.

Kind regards, 

on behalf of

Dr. Yashendra Sethi 

Academic Editor

PLOS ONE